# People overlook subtractive solutions to mental health problems
Tom J. Barry [1] ✉ & Nadia Adelina [1,2]

To solve problems, people tend to add new components to them rather than subtract from them. Across eight experimental and naturalistic studies, we examined if more additive (to start something new or do more) than subtractive advice (to stop or do less) is given when humans and artificial intelligence give mental health advice. Compared to subtractive advice, additive advice was recommended more frequently and was rated as more effective and feasible. Several moderators of this effect are explored: Older people are more additive than younger people, who we are advising (e.g., strangers/friends) matters, as does the type of pre-existing mental health-affecting activities (e.g., gambling versus avoiding exercise). People overlook subtractive advice when advising others and instead tell them to do more. Future research must explore the contribution of this additive advice bias to people's sense of being overwhelmed.

The prevalence of mental health problems and their costs to individuals and societies are increasing[1–3]. Many health services around the world are struggling to meet this growing need[4,5]. Even if they could, many people would still not engage with formal healthcare[6] and those that did would only receive modestly effective treatments[7]. Community-based solutions, such as help given by non-expert family or community group members[8], are necessary. To guide non-experts in supporting one another, we must first understand how they already support one another.

People talk to one another to attract socio-affective (e.g. validation) and cognitive (e.g. advice) support with the feelings and events that they experience[9]. Research in this area focuses predominantly on quantifying self-reported perceptions of the presence/absence or frequency of different support types (e.g. cognitive, etc.) and associating this with outcomes of interest (e.g. mental health symptoms, etc.)[10–12]. This overlooks what is actually said when people give support or advice (with few exceptions, e.g.[13]).

When recommending solutions to problems, people overlook solutions that involve the *subtraction* or removal of existing components[14,15]. Instead, people recommend *additive* changes wherein components are added to the original problem state. This is the case when solving basic geometrical problems and also in the recommendations people give to others to solve problems. For example, an analysis of recommendations given by incoming presidents to their new universities found that only 11% of these recommendations were subtractive[14].

We examine whether additive biases exist in advice about mental health (i.e. advice to do more or to take up new activities). This might explain why people feel time-poor and overwhelmed[16] and why the self-management of mental health can seem insurmountable[17]. We examine this experimentally where participants provide advice to hypothetical people presented as experiencing depressive or anxious symptoms (Studies 1, 2, 5–7). We also examine it naturalistically on Reddit where people advise strangers (Study 3). We examine whether effects are consistent across advice given to strangers, ourselves and our friends (Study 7) and we examine whether additive and subtractive advice are perceived differently in terms of their effectiveness, feasibility and benefit (Studies 4, 5–7). We also explore how the activities people are already doing which affect their mental health can in turn affect the advice that they are given (Studies 1, 2, 6.5, 7). Given the increasing popularity of chatbots (e.g. ChatGPT) for mental health advice[18] we also examine whether these bots exhibit similar preferences for additive advice (Study 8).

## Methods

The studies included here were approved by the ethics boards of the University of Hong Kong Human Research Ethics Committee (Study 1: EA210328; Study 3: EA220107) and the University of Bath Social Sciences Research Ethics Committee Ethics Committee (all other studies: 0095-268). All data, analysis scripts, study materials and coding guides are available on the Open Science Framework https://osf.io/edh8q/.

### Sample size determination

Sample estimation within Linear Mixed Models require data for simulation. As such, for Study 1 we estimated the required sample size per a similar mixed ANOVA with one within-subjects factor (Advice type: Additive, Subtractive) and a between-subjects factor (Gender: Man, Woman). A sample of 200 participants had 80% power to detect small-to-moderate effects ($f = 0.10$). We later used Study 1 data and the mixedpower()[19] $R$ package to determine the necessary sample size for the subsequent Linear

[1]Bath Anxiety and Mood (BAM) Research Group, Department of Psychology, University of Bath, Bath, UK. [2]Department of Psychology, University of Hong Kong, Hong Kong, China. ✉e-mail: tb2249@bath.ac.uk; tom.j.barry@icloud.com

Mixed Models used in our analyses. In these simulations (number of simulations = 1000), $N = 200$ had 100% power to detect the effect of advice type and 90.7% power to detect an interaction between advice type and gender. Our experience of Study 1 indicated that many potential participants would be excluded from analyses for not meeting our inclusion criteria given the nature of crowdsourced participant pools (e.g. a self-reported lack of seriousness or exclusions related to age or location). Even with attrition of 20% and 160 remaining participants, simulations indicated that our models would still have 100% power to detect the main effect and 85.8% power to detect interactions (e.g. with gender).

## Data processing, analysis and reporting

Inter-rater agreement was calculated using the codia[20] and irr[21] packages in R[22]. All analyses use lmerTest[23] in R to conduct Linear Mixed Models. Data were assumed to be normal but this was not formally tested. A fixed effect for advice type (additive, subtractive) was included in all models in all studies and, depending on the study, other fixed effects for participants gender and age and vignette gender and age, (Studies 1, 2, 4–8) and harm type (studies 6.5, 7, 8) were also added, as well as the interactions between these effects. For interpretability, results are reported using the anova() wrapper. Alpha was set to 0.05. Follow-up examinations of significant effects were conducted using emmeans()[24] with a Tukey adjustment for multiple comparisons, where appropriate.

## Study samples and procedures

**Study 1**. Study 1 examined whether participants gave more additive than subtractive advice in their open text responses to hypothetical vignettes of people with mental health difficulties when instructed to offer advice to improve their mental health. Study 1 was pre-registered on 18/09/23 here https://osf.io/y9vqz/.

Participants. 258 participants were recruited through MTurk via. Cloud-Connect research. Participants were paid the federal minimum wage in the United States ($7.25/hour) pro-rated for 15 min of study participation ($1.80). Participants were ineligible for participation if they were under the age of 18, had completed the survey once before, had participated in the preceding pilot, or were outside the U.S. In addition, 18 participants were excluded ($n = 1$, rated their English as 'Very Bad'; $n = 1$ did not identify as either man or woman; $n = 3$ did not complete the survey, $n = 13$ did not provide a codeable response). Data from 240 participants (Men = 125; Women = 115; Age, range = 19–76, $M = 41.83$, SD = 12.75) were analysed. For this and all other studies, participant gender was self-reported by participants.

Measures. Prior to Study 1, 200 separate participants (Men = 80; Women = 120; Age, range = 19–76 years, $M = 32.65$, SD = 11.28) on MTurk were asked 'What activities or lifestyle elements

do you currently engage in that you think influence your mental health, for better or

worse?'. These were then coded for activities that improve and worsen mental health and these codes were counted to deduce the most common activities within each category. From these, four vignettes were created that each included three symptoms of either Major Depressive Disorder (MDD) or Generalised Anxiety Disorder, as listed in the Diagnostic and Statistical Manual of Mental Disorders 5th Edition[25], and which referred to either (1) A woman engaging in a low (four) number of activities (2) A woman engaging in a high (eight) number of activities (3) A man engaging in a low number of activities (4) A man engaging in a high number of activities. Vignette gender was communicated within the vignettes using names selected from lists of the most common names for each gender. Each of the four vignettes included an activities that were the most frequently mentioned in the pilot study for improving and worsening mental health.

Procedure. After providing informed consent and demographic information, participants were presented each of the four vignettes in a random order and for each were asked 'What advice would you give to improve the current mental state of [name]?'.

Coding. A research student who worked on the project and a second independent coder who was blind to study aims and hypotheses double coded each response for the number of additive (*recommendations doing new or more of activities or lifestyle elements*) and subtractive pieces of advice included within. The coders first coded 300 responses and discussed any discrepancies and updated the coding manual with examples. After this the coders independently coded the remaining responses and average ICC was 0.96. Discrepancies were further resolved and analyses were conducted on these consensus codes.

**Study 2**. Study 2 examined whether Study 1 findings replicate using coders blind to study hypotheses. Study 2 was pre-registered on 14/09/23 here https://osf.io/7ksf5/.

Participants. Participants were sampled using the same platform and paid the same rate as in Study 1. 266 participants engaged with the survey initially but 62 were excluded per our inclusion criteria (as described in Study 1). 204 participants met the inclusion criteria (Men = 100; Women = 104; Age, range = 18–71 years, $M = 37.95$, SD = 11.45).

Measures. Chat GPT was given the Study 1 vignettes as examples and was asked to create new vignettes including three symptoms of Major Depressive Disorder or an anxiety disorder and to include in each three activities that might worsen mental health and three activities that might improve it (examples of each were provided). We asked it to create two vignettes representing each of young and old men and women, representing eight vignettes in total.

For each vignette, we asked participants to rate the relatability of the situation described in the vignette in 7-point Likert scale (1 = Not at all relatable; 7 = Highly relatable).

Participants were also asked 'What solutions would you offer to help Samuel manage his mental health?' (e.g. 'Samuel could participate in meditation and reduce his caffeine intake').

Procedure. After providing informed consent and demographic information, participants were shown the vignettes in a random order and for each were asked the question regarding the solutions they would offer, and then were asked about relatability. After this they were shown the debrief information and were redirected to the participation platform.

Coding. Two coders blind to study hypotheses and aims first read the coding guide from Study 1 and trained using four batches of 50 responses from Study 1. After each batch, the coders discussed codes and resolved disagreements. 20% of responses from Study 2 were then randomly selected and double coded (average ICC = 0.96). The remaining 80% were then split between coders and coded independently. The number of additive and subtractive responses for each level of vignette age and gender was averaged.

**Study 3**. Study 3 examined if earlier findings replicated in a naturalistic setting where people gave advice to people about their mental health on Reddit. Study 3 was pre-registered on 18/09/23, here https://osf.io/a4mzb/. Our pre-registered analysis plan included an analysis of whether responses that were gave more additive versus subtractive were assigned more or less comment karma/upvotes by other Reddit users. However, after scraping these data, these analyses were not possible given floor effects produced by the low amount of karma given to responses which prevented our models from converging.

Sample. The data were extracted as part of another study and the extraction process is described in detail there[26]. We used Python to extract all public posts on r/Anxiety and r/Depression March 1, 2021, and March 31, 2021 (a randomly selected month in 2021). 300 original posts from each subreddit

were then randomly sampled along with their first level comments. These samples were coded for whether or not they were advice seeking until 200 advice seeking posts were found. First level comments were then selected for coding if they gave advice (e.g. offered at least one solution), as many comments in Reddit only validate emotional experience (e.g. 'That must suck') and including responses with no advice would produce a large floor effect in our analysis. 121 responses were identified and selected for coding.

Coding. Two coders blind to study hypotheses coded all data. Coders read the coding manual from Study 2 and met to code 30 responses from Study 2 together. Next, they coded 160 responses from Study 2 separately and then came together to resolve disagreements. A further 160 responses from Study 2 were then coded (average ICC = 0.93). Coders then double coded 15% of the Reddit data. As ICC was satisfactory (average ICC = 0.8) the remaining data were split and independently coded.

Study 4. Study 4 examined whether additive advice is viewed as more effective and feasible than subtractive advice. Study 4 was pre-registered on 16/10/23 here https://osf.io/ychae/. Analyses of Study 4 data were pre-registered to include an analysis of vignette gender and the analyses reported here included this variable within the models. However, following Study 6.5, it was clear these analyses were confounded by the harm type described within the vignettes (see Study 6.5). We continue to include vignette gender within the models because of our pre-registered plan and to account for the variance explained by this confounding variable, but we do not present the effect of vignette gender so as to avoid suggesting that there were significant effects of vignette gender. Removal of the vignette gender variable from the models does not change any of the significant effects.

Participants. Participants were recruited via. CloudConnect with the same payment terms and inclusion/exclusion criteria as in previous studies. 218 people initially participated in the study. One person was excluded for residing outside of the US and another participant was excluded for not disclosing their gender. Per our pre-registered analysis plan, 24 participants were excluded from the analyses for incomplete data and two participants were excluded because they did not consent to analysis of their data after participating in the study. 190 participants were included in analyses (Men = 106; Women = 84; Age, range = 19–78 years, $M$ = 38.57, SD = 12.19).

Measures. As vignette age was not included as a factor in this study, participants were presented with only six of the eight vignettes present in Study 2, three vignettes from men and three from women. After each, participants were shown a response including four additive changes, one including four subtractive changes and one including two of each.

After each response, participants rated on scales from 1 (Strongly Disagree) to 7 (Strongly Agree) the extent of their agreement with each of seven statements. Three statements related to the effectiveness of the changes recommended in the response ('This solution would be an effective way to improve this person's mental health'; 'This solution may lead to quick improvements in this person's situation'; 'This solution may produce long-lasting improvement in this person's situation'; modified from the Abbreviated Acceptability Rating Profile[27]), the acceptability or feasibility of the changes ('The solution would be possible for this person to carry out'; 'The amount of time this solution would take is reasonable'; 'The amount of effort this solution would require is reasonable'; modified from the Behavioral Interventionist Satisfaction Survey[28] and the Structured Assessment of Feasability[29]) and their willing to use the changes recommended in the responses ('I would be willing to use this solution, if I were struggling'[27]). Average scores were computed for the three items related to each of effectiveness and acceptability.

Procedure. After providing informed consent, participants were presented the vignettes in a random order. For each vignette, the additive, subtractive and mixed responses were each presented in a random order. For each response option participants were presented with the rating scale. After completing the rating scale for a given response option, the next response option and its associated scales were shown. After all scales were completed, participants were asked to provide consent for the use of their data, after which they were redirected to CloudConnect for payment.

Study 5. Study 5 examined if earlier experimental findings replicate when participants are given explicit opportunities to be more subtractive in their advice. Study 5 was pre-registered on 22/03/2024 here https://osf.io/e7pv2. As in Study 4, vignette gender was included in our pre-registered but we do not present the findings for this effect given the confound of harm type.

Participants. Participants were recruited per the procedure outlined in previous studies and the same payment rates. 279 people initially participated in the study. 22 people were excluded for residing outside of the US and three participants were excluded for not disclosing their gender. 84 participants were excluded from the analyses for incomplete data. Five participants were excluded because they gave a score of 1 or 2 on a five-point seriousness scale (1 = Not at all serious; 5 = Very serious) at the end of the experiment (they were told this would not affect their payment). 161 participants were included in analyses (Men = 81; Women = 80; Age, range = 20–75 years, $M$ = 38.83, SD = 11.84).

Measures. Participants were presented with the same six vignettes as in Study 4. As in Study 1 and 2, the number of additive and subtractive changes recommended by participants for each vignette was scored and was averaged across each level of vignette gender. In addition, participants were asked to rank the changes that they suggested 'in terms of the extent to which you think they will benefit the person's mental health'. From this the number of additive and subtractive changes that appeared in the top five most beneficial recommendations were counted for each vignette and averaged across levels of vignette gender. We also measured relatability using the question from Study 2.

Procedure. After providing informed consent, participants were given information about what subtractive and additive changes were, they were shown a practice vignette 'Carl has depression, he swims regularly and smokes.' and were asked to provide one subtractive change in the corresponding box and one additive change in the box below that. They were then told what example correct answers were (quit smoking; take up yoga). After this, participants were shown each of the test vignettes in a random order. After each they were given five boxes in which to enter subtractive changes and then five boxes to enter additive changes. Participants did not need to enter text into every box. After they had entered all of their recommended changes, participants were asked to rank them from most to least beneficial, after which they answered the relatability question. After all vignettes had been shown participants were debriefed and then rated their seriousness and gave final consent to use of their data.

Study 6. Study 6 was a replication of Study 5 with additional counter-balancing. Study 6 was pre-registered on 15/04/2024, here https://osf.io/t5b64/. As in Studies 4 and 5, vignette gender was included in our pre-registered but we do not present the findings for this effect given the confound of harm type.

Participants. Participants were recruited as in previous studies and were paid the same rates. 276 people initially participated in the study. Five people were excluded for residing outside of the US and five participants were excluded for not disclosing their gender and eight because they did not disclose their age. 77 participants were excluded from the analyses for incomplete data. Five participants were excluded because they reported that they were not serious. 176 participants were included in analyses (Men = 84; Women = 92; Age, range = 20–80 years, $M$ = 39.24, SD = 13.2).

Measures and Procedure. Study 6 used the same measures and procedure as Study 5, however, the vignettes were counterbalanced such that the male vignettes now used the female names and vice versa.

**Study 6.5.** Study 6.5 was a re-analysis of Study 5 and 6 data where the data were recoded for *harm* to explore whether the harms the person with the mental health difficulties was described as engaging in influenced the kinds of advice that they were offered. Study 6.5 was pre-registered on 16/07/2024 here https://osf.io/g5xa4/.

Data. The number of activities within each vignette that were likely to be directly harmful were counted (*positive harms*; e.g. smoking) and as were the number of activities that were likely to be harmful because the person is described as not doing something healthy (*negative harms*; e.g. not going for walks). This suggested that two of the six vignettes had two positive harms and one negative and two others had two negative harms and one positive harm. These vignettes were selected from both data sets and represented as positive and negative harm conditions, respectively, within the analyses.

**Study 7.** Study 7 examined harm type on advice giving in a set of new vignettes and also explored whether advice differed based on whether it was offered to a stranger, one's self or a friend. Study 7 was pre-registered on 01/06/2024 here https://osf.io/3zbe5.

Participants. Participants were recruited as before and received the same payment. 253 people initially participated in the study. One person was excluded for not disclosing their country of residence, three participants were excluded for not disclosing their gender and three because they did not disclose their age. 52 participants were excluded from the analyses for incomplete data. Four participants reported that they were not serious. 190 participants were included in analyses (Men = 102; Women = 88; Age, range = 19–68 years, $M = 32.98$, SD = 10.72).

Measures. Study 7 included the same dependent variables as studies 5 and 6. The vignettes were adapted from the four selected in study 6.5. The same gendered names were used but the vignettes were changed to include either three positive or three negative harms. As such, the vignettes no longer included any activities that were previously included as potentially beneficial to mental health. The vignettes were fully counterbalanced for gender and harm such that 16 vignettes were created, comprising four counterbalance conditions. In addition to the vignettes, participants were told 'Please think about your own mental health: Think about the things you currently do that you think affect your mental health and the changes they could make that might have a positive impact on your mental health.' They were then asked. 'What you would recommend to yourself to improve your mental health?', after which they could provide subtractive and then additive changes—as in studies 5 and 6—and then rank their changes. Participants were also asked 'Please think of a close friend's mental health…' followed by the same instructions, with adjustment of pronouns.

Procedure. Study 7 used the same procedure as studies 5 and 6 with the addition that the study was conducted in two blocks. In the first block, participants were asked about themselves and a close friend, presented in a randomised order. In the second block, participants were randomly allocated to one of the four counterbalance conditions and the four vignettes were then presented in a random order. After this, participants were debriefed and answered seriousness and final consent questions.

**Study 8.** Study 8 examined whether GPT is more additive than subtractive in the advice that it gives when people ask it to help them with their mental health. Study 8 was pre-registered on 15/05/2024 here https://osf.io/2ja8g/.

Sample. There was no established precedent for estimating sample size using GPT as a 'participant'. As such, we used the sample size estimates from our

other studies. To extract an equivalent number of responses as we would for human participants, where each of $N = 200$ responded to four vignettes, we extracted $n = 50$ per each of the 16 vignettes from Study 7 (given the $2 \times 2$ design and the full counterbalancing therein).

Procedure. The GPT API was accessed using httr()[30] in R. For each vignettes, GPT received the following prompt (with temperature set to 0.7), encouraging it to respond as a participant in our other studies might: 'You are an American adult over 18. I'm going to show you four different descriptions of peoples' mental health. After each description I will ask you to recommend some changes the person could make to help them manage their mental health. What solutions would you offer to help this person manage their mental health?'

Coding. Two coders blind to the study hypotheses in a staged manner where they were trained to code the data while coding it after reading the coding manual from Study 7. In the first three stages, the coders coded the same data separately, agreement was calculated and then disagreements were resolved. In the final stage the remaining data were split and the coders worked independently. They first coded 25 responses (average ICC = 0.84), then 75 responses (average ICC = 0.97) and then 100 (average ICC = 0.84). As ICC was consistently high they then coded the rest of the data independently.

### Reporting summary
Further information on research design is available in the Nature Portfolio Reporting Summary linked to this article.

### Results
Figure 1 includes violin plots of additive versus subtractive advice given across studies. Full statistical reporting of all effects is available in the online supplement.

### Advice in response to hypothetical mental health difficulties
In Study 1, participants were shown accounts of people experiencing mental health difficulties with information on what activities in their life were affecting their mental health. Participants gave open text advice to each account to help them improve their mental health and this advice was coded for the amount of additive versus subtractive advice given. Participants offered more additive (EMMean = 2.801; SE = 0.060) than subtractive advice (EMMean = 0.931; SE = 0.060), $F(1, 1652) = 63.061$, $p < 0.001$, partR$^2$ = 0.005, 95% CI[0.013, 0.001].

Study 2 used the same design to examine if Study 1 findings replicated. Participants suggested more additive (EMMean = 1.516; SE = 0.029) than subtractive advice (EMMean = 0.447; SE = 0.029) in their open text responses to accounts of mental health difficulties, $F(1, 1419.12) = 1291.66$, $p < 0.001$, partR$^2$ = 0.073, 95% CI [0.088, 0.051]. Again, although there was an interaction between the amount of additive versus subtractive advice offered and participant gender, $F(1, 1419.12) = 22.592$, $p < 0.001$, partR$^2$ = 0.006, 95% CI[0.016, 0.001], both men, $b = 0.927$, SE = 0.043, $t(1420) = 21.846$, $p < 0.001$, 95% CI[0.844, 1.010], and women, $b = 1.210$, $SE = 0.042$, $t(1421) = 29.051$, $p < 0.001$, 95% CI[1.128, 1.290], suggested more additive than subtractive solutions. There was also an interaction between the amount of advice offered of each type and participant age, $F(1, 1418.90) = 4.7202$, $p = 0.030$, partR$^2$ = 0.003, 95% CI[0.011, 0.000], such that participants became more additive the older they were, $b = 0.006$, SE = 0.003, $t(365.58) = 2.304$, $p = 0.022$, 95% CI[0.001, 0.012], but there was no statistically significant association between age and subtractive advice, $b = 0.000$, SE = 0.003, $t(365.22) = 0.107$, $p = 0.915$, 95% CI[−0.005, 0.006].

### Naturalistic advice-giving on Reddit
In Study 3, we examine whether the preference for additive advice giving occurs in naturalistic settings. Advice-seeking posts and their associated comments from r/Depression and r/Anxiety on Reddit were scraped. Comments were coded for the amount of additive and subtractive advice they gave. Users offered more additive (EMMean = 1.860; SE = 0.113) than

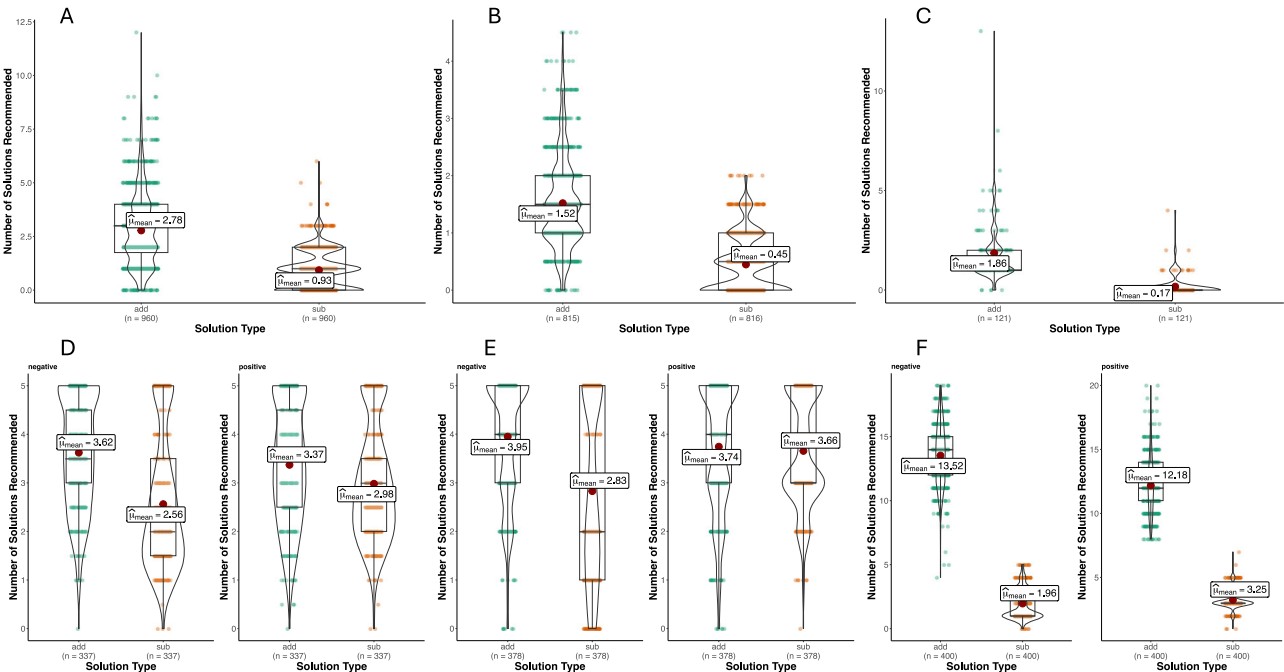

**Fig. 1 | Amount of additive versus subtractive advice.** Note. Number of additive and subtractive pieces of advice given in response to hypothetical vignettes (Study 1, **A**; Study 2, **B**), on Reddit (Study 3, **C**) and in response to vignettes that emphasised the engagement in activities that were harmful for mental health by their presence ('positive', e.g. gambling) or through their absence ('negative', e.g. not exercising) (Study 6.5, **D**; Study 7, **E**; Study 8, **F**).

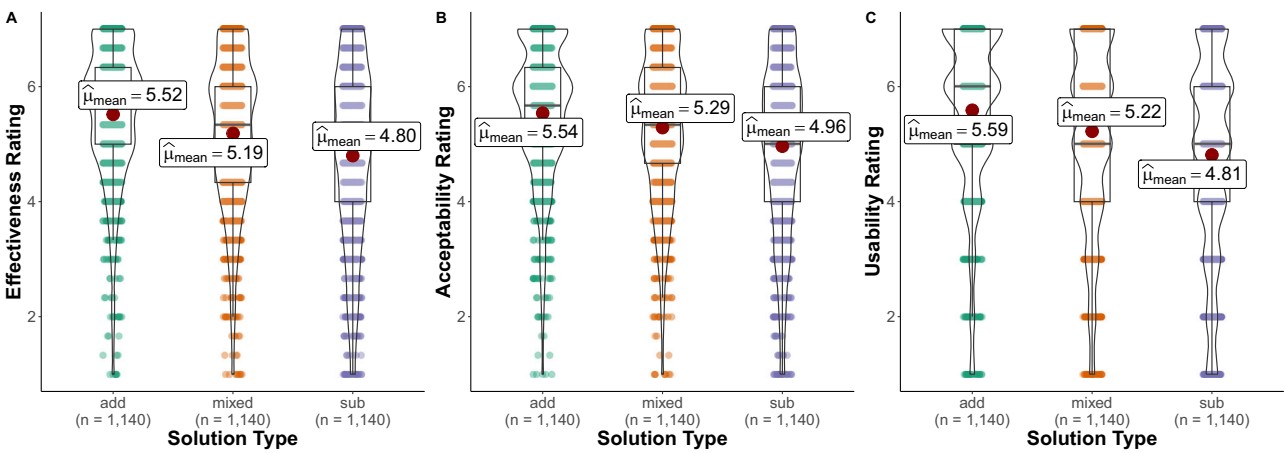

**Fig. 2 | Effectiveness, acceptability and usability of advice.** Note. Self-reported ratings of effectiveness (**A**) and acceptability (**B**) of, and willingness to use (**C**), solutions that are purely additive, subtractive or a mixture of the two (Study 4).

subtractive advice (EMMean = 0.174; SE = 0.113), $F(1, 119) = 80.837$, $p < .001$, partR$^2$ = 0.229, 95% CI[0.320, 0.146]. There was no statistically significant main effect of the number of activities a person reported themselves as already engaged in, $F(1, 119) = 0.000$, $p = 0.993$, partR$^2$ = 0.000, 95% CI[0.022, 0.000], or interaction of this with Solution, $F(1, 119) = 0.103$, $p = 0.749$, partR$^2$ = 0.000, 95% CI[0.022, 0.000].

### Effectiveness and feasibility

Study 4 examined whether additive advice is given more frequently than subtractive advice because it is viewed as more effective or feasible. Participants were shown the cases of mental health difficulties, and were then shown three different advice options (entirely additive, entirely subtractive or mixed) and asked to rate the effectiveness and feasibility of each. Participants rated entirely additive advice as being more effective than entirely subtractive advice, $b = 0.732$, SE = 0.038, $t(2840) = 19.497$, $p < 0.001$, 95% CI[0.644, 0.821], and advice that was a mixture of additive and subtractive,

$b = 0.338$, SE = 0.038, $t(2840) = 9.003$, $p < 0.001$, 95% CI[0.250, 0.426]. The additive advice was also rated as more acceptable or feasible than subtractive, $b = 0.594$, SE = 0.038, $t(2840) = 15.451$, $p < 0.001$, 95% CI[0.504, 0.684], and mixed advice, $b = 0.259$, SE = 0.038, $t(2840) = 6.727$, $p < 0.001$, 95% CI[0.168, 0.349]. Participants also said that if they experienced the same difficulty as the person depicted in the scenario that they would be more likely to adopt the additive advice than the subtractive, $b = 0.800$, SE = 0.049, $t(2840) = 16.233$, $p < 0.001$, 95% CI[0.684, 0.915] or mixed advice, $b = 0.376$, SE = 0.049, $t(2840) = 7.642$, $p < 0.001$, 95% CI[0.261, 0.492](See Fig. 2).

### Emphasising subtractive advice

Studies 5 and 6 presented hypothetical accounts (as in studies 1 and 2) and gave participants information about additive and subtractive advice, gave examples of each and gave them practice giving each. It also guided participants towards giving five of each advice type. Participants also ranked the advice they gave for their potential benefit. In Study 5, participants still gave

more additive (EMMean = 3.290; SE = 0.085) than subtractive advice (EMMean = 2.690; SE = 0.085), $F(1, 472.69) = 174.722$, $p < 0.001$, partR$^2$ = 0.009, 95% CI[0.029, 0.000], even though they were explicitly taught about additive and subtractive responses, practiced both types, and were prompted to give (up to five) subtractive advice first. When asked to rank each piece of advice in terms of their potential benefit, participants also ranked more additive (EMMean = 2.44; SE = 0.042) than subtractive (EMMean = 1.93; SE = 0.042) advice in their top five most beneficial pieces of advice, $F(1, 447.42) = 75.414$, $p < 0.001$, partR$^2$ = 0.014, 95% CI[0.039, 0.002]

Study 6 replicated these effects with participants again offering more additive (*EMMean* = 3.57; *SE* = .086) than subtractive advice (EMMean = 2.91; SE = 0.086), $F(1, 517.07) = 235.018$, $p < 0.001$, partR$^2$ = 0.051, 95% CI[0.087, 0.024], and ranking more additive (EMMean = 2.45; SE = 0.042) than subtractive (EMMean = 2.17; SE = 0.042) advice in their top five most beneficial, $F(1, 667) = 22.652$, $p < 0.001$, partR$^2$ = 0.103, 95% CI[0.149, 0.065].

### Pre-existing activities and their harm

Study 6.5 recoded the scenarios from studies 5 and 6 to create a new variable capturing whether the advisee was described as engaging in mostly positive harms (i.e. activities that were harmful by virtue of their presence, e.g. smoking)

or negative harms (i.e. activities that were harmful by virtue of the absence of benefit, e.g. not exercising).

Even with this new factor included with analyses, there was still a main effect of advice type, $F(1, 998.73) = 395.17$, $p < 0.001$, partR$^2$ = 0.058, 95% CI[0.084, 0.036]. There was also an interaction between advice type and harm type, $F(1, 998.73) = 86.156$, $p < 0.001$, partR$^2$ = 0.022, 95% CI [0.022, 0.002], but participants suggested more additive than subtractive solutions irrespective of whether people were engaged in predominately in negative, $b = 1.063$, SE = 0.052, $t(998) = 20.605$, $p < 0.001$, 95% CI[0.962, 1.164], or positive harms, $b = 0.386$, SE = 0.052, $t(998) = 7.498$, $p < 0.001$, 95% CI[0.285, 0.487] (See Fig. 1). The same was also true in the analysis of benefit rankings where the main effect of advice type was present, $F(1, 1278) = 116.655$, $p < 0.001$, partR$^2$ = 0.105, 95% CI[0.137, 0.076], and the interaction with harm type, $F(1, 1278) = 46.205$, $p < 0.001$, partR$^2$ = 0.020, 95% CI[0.038, 0.008]. Additive advice was ranked as more beneficial than subtractive advice irrespective of whether for participants were advising people predominantly engaged in positive harms, $b = 0.186$, SE = 0.066, $t(958) = 2.831$, $p = 0.005$, 95% CI[0.057, 0.315], or negative harms, $b = 0.817$, SE = 0.066, $t(958) = 12.444$, $p < 0.001$, 95% CI[0.688, 0.946].

Study 7 further examined the effects of prior engagement with types of harmful activities by showing participants scenarios of people either exclusively engaged in positive harms or negative harms. Again, participants were asked to provide advice and then to rank this advice for anticipated benefit. In addition to replicating the main effect of advice type present in other studies, $F(1, 938.02) = 42.810$, $p < 0.001$, partR$^2$ = 0.000, 95% CI[0.006, 0.000], there was also a significant interaction between advice and harm type, $F(1, 1298.16) = 166.268$, $p < 0.001$, partR$^2$ = 0.007, 95% CI[0.019, 0.001]. Participants suggested more additive than subtractive advice to people who were engaged in negative harms, $b = 1.152$, SE = 0.058, $t(1298) = 19.880$, $p < 0.001$, 95% CI[1.039, 1.266], but there was no statistically significant difference in the number of additive and subtractive solutions suggested to people who were engaged in positive harms, $b = 0.098$, SE = 0.058, $t(1298) = 1.690$, $p = 0.091$, 95% CI[−0.016, 0.211]. The same was true for benefit rankings with additive advice being ranked as significantly more beneficial than subtractive advice for people engaged exclusively in negative harms, $b = 1.030$, SE = 0.086, $t(1269) = 12.035$, $p < 0.001$, 95% CI[0.862, 1.198], but there was no significant difference for people engaged exclusively in positive harms, $b = −0.126$, SE = 0.085, $t(1269) = −1.478$, $p = 0.140$, 95% CI[−0.293, 0.041].

### The importance of who is being advised

In Study 7, in addition to advising strangers (as in previous studies) participants were also asked to think about themselves and a close friend and to advise each on how to improve their mental health. The amount of additive versus subtractive advice participants gave differed based on whether it was given to a stranger, ourselves or a friend, $F(2, 938.02) = 17.344$, $p < 0.001$, partR$^2$ = 0.006, 95% CI[0.018, 0.000]. There was no statistically significant difference between the amount of additive and subtractive advice recommended to close friends, $b = 0.049$, SE = 0.073, $t(938) = 0.667$, $p = 0.505$, 95% CI[−0.094, 0.192], but people were more additive than subtractive when advising strangers, $b = 0.620$, SE = 0.073, $t(938) = 8.500$, $p < 0.001$, 95% CI[0.477, 0.764], and themselves, $b = 0.157$, SE = 0.073, $t(938) = 2.155$, $p = 0.031$, 95% CI[0.014, 0.300]. People were significantly more subtractive when advising their friends, $b = 0.691$, SE = 0.073, $t(938) = 9.474$, $p < 0.001$, 95% CI[0.520, 0.862], and themselves, $b = 0.704$, SE = 0.073, $t(938) = 9.655$, $p < 0.001$, 95% CI[0.875, 0.533], than they were when advising strangers. Subtractive solutions were also ranked as more beneficial than additive solutions when advising friends, $b = −0.368$, SE = 0.113, $t(903) = −3.262$, $p = 0.001$, 95% CI[−0.590, −0.147], whereas when advising strangers, additive solutions were rated as more effective than subtractive solutions, $b = 0.438$, SE = 0.113, $t(903) = 3.876$, $p < 0.001$, 95% CI[0.216, 0.660]. There was no statistically significant difference between solution types when advising oneself, $b = −0.104$, SE = 0.113, $t(903) = −0.924$, $p = 0.356$, 95% CI[−0.325, 0.117].

### Advice-giving by ChatGPT

In Study 8, vignettes from previous studies were given to GPT 4o along with the same instructions as previous studies to provide advice to improve mental health. GPT responded similarly to participants in other studies in that it was more additive than subtractive, $F(1, 796.01) = 13139.938$, $p < 0.001$, partR$^2$ = 0.703, 95% CI [0.722, 0.683]. There was evidence of differences in these effects based on vignette gender, $F(1, 796.01) = 4.696$, $p = 0.031$, partR$^2$ = 0.004, 95% CI [0.013, 0.000], and the harm type that the person in the vignette was said to be engaged in, $F(1, 796.01) = 215.912$, $p < 0.001$, partR$^2$ = 0.050, 95% CI [0.072, 0.031] (Fig. 1). However, irrespective of the levels of these potential moderating variables, significantly more additive solutions were offered than subtractive solutions (all p's < 0.001 within paired contrasts).

### Discussion

People offered significantly more additive advice (e.g. exercise more) than subtractive advice (e.g. quit gambling). This tendency persisted across multiple contexts: when participants provided open-text responses (Studies 1, 2), on Reddit (Study 3), and even under conditions designed to elicit subtractive advice (e.g. explicit instructions differentiating between subtractive and additive advice, examples of each advice type, prompts to give subtractive advice first, and equal space to provide five of each) (Studies 5, 6, 7). ChatGPT was also more additive than subtractive (Study 8).

We therefore possess a reliable tendency to give more additive than subtractive advice for mental health problems. This bias has the potential to create a social context in which we are advising one another to always do more. Several explanations for this bias have been proposed[31]. As the neural systems that underlie our understanding of negative value develop later than those that underlie our understanding of positive value this may create an enduring advantage for addition over subtraction. Alternatively, humans also possess a bias towards tangible solutions in decision making[32,33] and it may be that additive changes are more tangible than subtractive changes given their more obvious presence once implemented.

Gender emerged as a possible moderator of our observed effects but men and women were both more additive than subtractive (Studies 1, 2, 5, 6, 7) and we did not find evidence that this tendency differed as a function of whether they were advising men or women (see Study 7 in supplemental analyses). Participants were more additive the older they were (Studies 1, 2) and this did not differ as a function of whether they were advising an old or

young person (Study 2). This may indicate that any developmental advantage given to additive, relative to subtractive, processing may increase as we age[34].

There was evidence that who we are advising matters. Participants were more additive than subtractive when advising strangers and themselves but not a close friend (Study 7). It is possible that participants did not know enough about the strangers to offer equivalent amounts of additive and subtractive advice. However, people were more additive than subtractive when advising themselves. Also, across the studies participants rarely subtracted the three harmful activities that the person in the vignette was said to be engaged in. In studies 5 and 6 participants were also implicitly encouraged to give five pieces of subtractive advice before giving any additive advice and yet they still responded in the expected direction. Participants also rated additive advice as significantly more effective and feasible (Study 4) and as having greater potential benefit (Studies 5, 6 and 7) than subtractive advice.

Although the contrast between additive and subtractive advice was significant when advising oneself, the data in Fig. 3 and the regression coefficient for this contrast indicate that this effect is small. Participants were significantly more subtractive to themselves than to strangers. Again, this could perhaps have been because people know more about themselves than the strangers, so it is easier to recommend subtractive advice. It is of note that this was within our study design that encourages subtractive responses by giving participants five spaces to give subtractive advice before they give additive advice. It is possible that, as with our other effects, this effect might be larger in other designs that allow open text advice giving. It is also of note, however, that the interaction between advice type and advice recipient was evident alongside the main effect of advice type. Irrespective of the advice recipient, the advice we give is markedly additive.

Although the number of activities a person was engaged in did not affect the advice they received (Study 1), the *type* of activity did affect their advice. Whether a person was engaged in activities that were detrimental to mental health through their presence (*positive harms*; e.g. gambling) or through the absence of benefit (*negative harms*; e.g. not exercising) affected advice (Studies 6.5 and 7). People described as engaging *solely* in positive harms received the same amount of additive and subtractive advice whereas people described as engaging solely in negative harms received significantly more additive than subtractive advice (Study 7). However, when people are described as engaging in *proportionally more* positive versus negative harms, or vice versa, both groups received more additive than subtractive advice (Study 6.5). Nevertheless, the findings indicate that people are inclined to subtract positive harms (e.g. gamble less) and offer additive solutions for negative harms (e.g. do more exercise). This effect was not replicated by GPT, which was always more additive than subtractive perhaps because it tended to give many pieces of additive advice, more than any human participants gave. For human participants, mental health literacy—or knowledge about the factors that can be of positive and negative harm to mental health and the activities that might subsequently be beneficial to a person – is likely to be an important moderator of these effects that should be explored in future research.

Nevertheless, combining the harm type findings with the findings regarding advice recipient, it may be that people offer similar amounts of subtractive and additive advice to our friends because we are more willing to subtract our friend's positive harms than our own. For example, we may be more willing to recommend that our friends quit gambling than we would recommend this to ourselves. Indeed, subtractive advice was ranked as more beneficial to our friends than additive advice. Conversely, participants

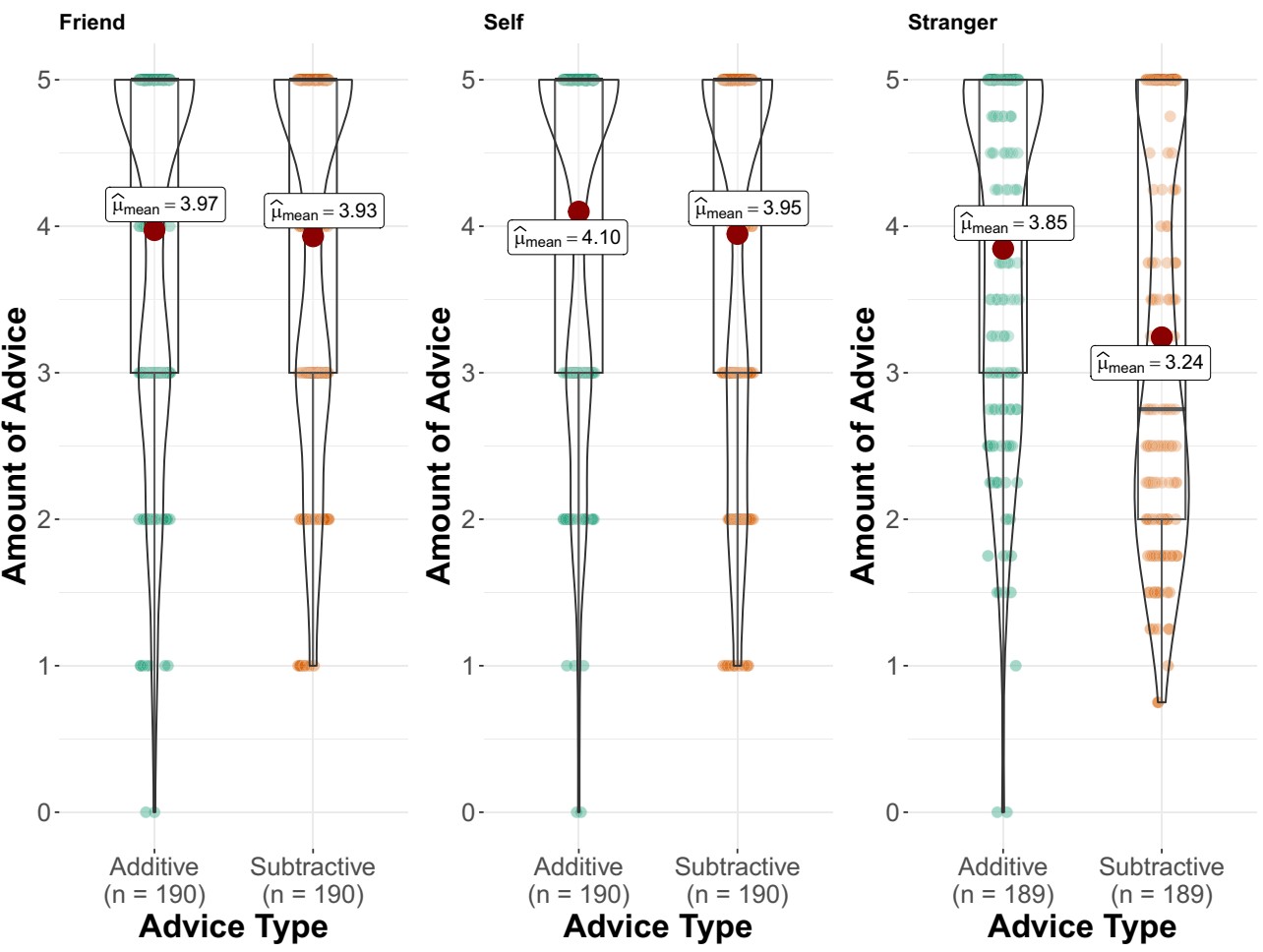

**Fig. 3 | Advice given to friends, strangers and self.** Note. The amount of additive and subtractive advice offered to friends, others/strangers and to oneself (Study 7).

reported that they would be more likely to use additive advice for themselves than subtractive advice.

Finally, GPT replicated the additive bias (Study 8), unsurprisingly given that GPT is trained on Reddit data where we observed the same effect (Study 3). As systems such as GPT become increasingly common sources of mental health advice[18] it is essential that they do not exacerbate existing advice-giving biases. These systems should be weighted to suggest activities that people might subtract from their life or to enquire for more information about existing activities people are engaging in or avoiding where they feel they do not have enough information to offer people subtractive guidance.

## Limitations

It is of note that, in the initial studies presented here, the vignettes included some details about activities that the person in the vignette preferred to do to help their mental health. This was intended to give participants some content upon which to base their advice. It is possible this may have biased participants towards recommending additive advice in line with these preferences. However, as can be seen in Fig. 1, the average amount of additive advice given in the studies often exceeded the amount of these details that were present in the vignettes. In addition, these details were removed in Study 7, and they were not present in the Reddit posts analysed in Study 3, and yet the bias towards additive advice was present in both.

It is also worth noting that the research presented here does not explore whether additive and subtractive advice differ in how beneficial or burdensome they are, once enacted. It remains possible that additive advice is indeed more beneficial than subtractive advice, however, intuitively it would seem that there are surely limits to how much additive advice one can follow. Also, our analyses, and those of other studies in this area[14,15], consider additive and subtractive advice as binary categories but it is possible that activities are more or less additive/subtractive, perhaps as a function of the effort required to enact them. Future research could explore this, perhaps within an Ecological Momentary Assessment design where advice benefit and burden are measured in the contexts in which advice adoption occurs.

## Conclusions

In a world where we already feel like we are time poor and doing too much, there is a sense that we must do yet more to cope with the sadnesses and anxieties of life. This might be compounded by advice to others that is predominantly telling them to do more. Future research should now explore the real-life burden of this additive advice bias.

## Data availability

All data are available here https://osf.io/edh8q/.

## Code availability

All materials and script are available here https://osf.io/edh8q/.

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

## Acknowledgements

This research was supported by the Research Promotion Fund of the Department of Psychology, University of Bath. The funders had no role in study design, data collection and analysis, decision to publish or preparation of the manuscript. This work was supported through the volunteer work of students from the University of Bath and University of Hong Kong: Jhalak Sheth, Paige Stephens, Mattias Weedman, Isabel Castelow, Mia Beastall, Roxsanne Patel and Mia Hazeltine.

## Author contributions

T.J.B.: Conceptualisation, Data curation, Formal Analysis, Funding acquisition, Investigation, Methodology, Supervision, Writing—original draft, Writing—review & editing; N.A.: Conceptualisation, Investigation, Methodology, Writing—original draft, Writing—review & editing.

## Competing interests

The authors declare no competing interests.
