## [Transparent Peer Review file · Communications Psychology]

People overlook subtractive solutions to mental health problems.

Corresponding Author: Dr Tom Barry

Version 0:

Decision Letter:

Dear Dr Barry,

Thank you for your patience during the peer-review process. Your manuscript titled "People overlook subtractive solutions to mental health problems." has now been seen by 2 reviewers, and I include their comments at the end of this message. They find your work of interest but raised some important points. We are interested in the possibility of publishing your study in Communications Psychology, but would like to consider your responses to these concerns and assess a revised manuscript before we make a final decision on publication.

We therefore invite you to revise and resubmit your manuscript, along with a point-by-point response to the reviewers. Please highlight all changes in the manuscript text file.

Editorially, we consider it important that the revised manuscript more thoroughly address the study limitations and situate the finding in the literature whilst staying close to what is shown by the data. Please rerun analyses including both vignette and participant as random effects. Preregistration links for each study as well as dates of preregistration must be included in the Methods section. Any deviations from the preregistrations should be reported. You can find our full policy on preregistration here: <https://www.nature.com/commspsychol/submit/preregistration>

Please ensure you follow our statistical guidelines when reporting statistics (<https://www.nature.com/commspsychol/submit/submission-guidelines#statistical-guidelines>). Please note in particular our requirements for the reporting and interpretation of null-results. Non-significant findings derived from null-hypotheses significance tests should be reported in full, but may not be interpreted. Where you interpret null results, this interpretation must be based on Bayes Factors or equivalence tests.

I am attaching an Editorial Requests Table that details critical reporting requirements for the revised manuscript. Please attend to each item and ensure your manuscript is fully compliant. If your revised manuscript is not aligned with these requests on major issues, such as those concerning statistics, it may be returned to you for further revisions without re-review.

Please submit the following items:

- Revised manuscript
- Point-by-point response to the referees' comments
- Cover letter (as a separate document)
- [Nature Research Reporting Summary](https://www.nature.com/documents/nr-reporting-summary.pdf)

- Completed Editorial Request Table (attached).

via this link: Link Redacted .

Additional guidance is available in our style and formatting guide Communications Psychology formatting guide.

Best regards,

Jennifer Bellingtier

Jennifer Bellingtier, PhD
Senior Editor
Communications Psychology

REVIEWER EXPERTISE:

Reviewer #2 public mental health, psychiatry and psychotherapy, mental health counseling
Reviewer #4 population mental health, mental health counseling

REVIEWER REPORTS:

Reviewer #2 (Remarks to the Author):

Thank you for letting me see this revised ms that I had commented upon for Nature Hum Behav. I am not particularly convinced by the authors' replies.

1. If you look at Figure 3, there is nothing magical about a p-value. The intuitive finding is that you have one pattern for friend and self - and a very different pattern for other. This is the difference that I would like to see reported and discussed.
2. Even if all participants were US based, information on their immigrant status (first and/or second generation) could still be analyzed to examine cultural differences.
3. I believe that MHFA has a mostly additive agenda. It does not work, MHFA "aid recipients" are not better off (Forthal et al 2022; Richardson et al 2023, Cochrane). Still, MHFA is sold worldwide, a corrupt endeavor. If the authors agree with this interpretation, why do they not mention it? Their research should have a real-world impact.

Reviewer #4 (Remarks to the Author):

The paper describes eight sequential studies broadly aiming to compare the frequency of additive (one should take an additional action) and subtractive (one should stop an action) mental health advice. The studies explored: whether individuals provide more additive or subtractive advice in response to vignettes of individuals experiencing mental health problems (Studies 1 and 2); whether Redditors give other users asking for mental health help more additive or subtractive advice (Study 3); whether individuals rate additive or subtractive advice as more effective, acceptable and feasible (Study 4); whether individuals provide more additive or subtractive advice in response to vignettes of individuals experiencing mental health problems after being prompted to give both subtractive and additive advice, and asked them to rank their advice in terms of benefit (Study 5 and Study 6); whether the likelihood of the advice in Study 5/6 to be additive or subtractive differed by harm type (Study 6.5); whether advice was more likely to be additive or subtractive based on the harm type that the

vignette depicted, and asked them to rank their advice in terms of benefit (Study 7); and, whether ChatGPT provided more additive or subtractive advice in response to the abovementioned mental health vignette (Study 8). The study is based on the paper “People systematically overlook subtractive changes” (Adams et al., 2021)

Broadly these studies found that individuals, Redditors, and ChatGPT give more additive than subtractive advice, aligning with evidence of a human bias for additive rather than subtractive changes found in Adams et al. (2021) and the Klotz book “Subtract: The Untapped Science of Less”. Within this context, it would be of interest to others in the field, describing this bias in the setting of laypersons’ mental health advice – a key area of intervention in mental health promotion. While the claims that individuals are more likely to both give additive advice, and perceive it as beneficial compared to subtractive advice, are convincing, the paper does not establish that additive advice in the context of the study’s eight experiments is a less effective, or more burdensome/stressful type of mental health advice than subtractive advice.

The methodological detail is fairly sound but there are several opportunities for bias in these methods that are not acknowledged by the manuscript. The development of the vignettes required binary coding of activities that may improve or worsen mental health (main manuscript, pg. 15), and the subsequent coding of the advice given by participants to these vignettes are also binary; as additive or subtractive advice (main manuscript, pg. 16). This assumes a clear distinction between these concepts, however in practice they can be mixed or ambiguous (e.g., “losing weight” might improve body image and mental health, or be a source of stress and poor body image). This could lead to coding being subjective, limiting replicability of this process.

I also wonder, having looked at the vignettes used by the study in OSF (<https://osf.io/vucbr>), if they bias participants towards giving additive advice by including suggestions of additive activities as being desired by the individual in the vignette. For example, “Vignette 2 Sarah noticed that she has been feeling overwhelmingly tense, restless and agitated. She finds herself worrying excessively about her job security, completing her errands on time, her health and many other aspects of her daily routine. Sarah often ruminates on the past, obsessing over the significance of someone’s actions. Sarah has expressed that her poor drinking habits have contributed to her current mental state. As well as, spending hours watching movies and tv shows. However, Sarah feels exercising daily could significantly improve her mental state. Engaging her creative side by painting could also lift her mood.” This vignette seems to suggest that Sarah believes that increasing her exercise and engaging in painting would improve her mental health, encouraging study participants to offer her this additive advice. This is perhaps a limitation to acknowledge in the discussion.

Regarding the statistical analysis, I noticed that vignette ID was included as random effect in Study 4 but not in the other studies (supplementary material, pg. 3). The design of the other studies suggest that each participant is shown multiple vignettes, and advice is given on each vignette by multiple participants, so why was the vignette ID not included as a random effect in these other analyses? Adding vignette ID as a random effect to the LMMs for the other studies could also mitigate bias from the tendency for some vignettes to elicit more additive advice.

I feel the claims are not entirely appropriately discussed in the context of the literature or findings. The study establishes through a range of studies that additive advice is more commonly given and rated as more beneficial. However, this is very distinct from the claims in the Abstract that “People feel that they are overburdened and yet we often tell them to do more. When helping people with their mental health, advice givers must balance what can be added to life with what needs to be removed” (main manuscript, pg. 2) and in the Discussion that “This has the potential to create a social context in which we are advising one another to always do more, leading to a worsening of perceived stress or burden” (main manuscript, pg. 9). The authors do not cite empirical literature linking additive advice to increased stress or burden. These claims are also not directly supported by the studies; indeed Study 4 participants judged the additive advice as more acceptable/feasible and were more likely to adopt it themselves. Furthermore, the additive and subtractive advice coded in the studies are sometimes semantically similar recommendations that may be functionally indistinguishable to the advisee; for example “I would suggest he improve his diet” is coded as additive while “she should cut her diet to be more healthy” is coded as subtractive (<https://osf.io/v3h9g>).

Thus, while the studies evidence a bias towards additive advice, the manuscript would benefit from a more cautious discussion of their implications. The conclusion that non-experts must consider “what we can do less of” may be true, but is not established by the studies and should be framed as a hypothesis for future research rather than a direct consequence of the findings.

While for the reasons described above I am not sure the manuscript is acceptable in its current form, it has the potential to be worthy of publication. My recommendations for the manuscript would be to acknowledge in the potential biases described above and evidence the idea of additive advice as burdensome with literature (or as a future research question) rather than as a finding of the research. There are some small spelling errors (“improve” should be “improve” on pg. 4, main manuscript) and some sentences are a bit confusing (“gave participants instructions about what additive and subtractive advice where and gave them practice giving each.” on pg. 6, main manuscript) that would be worth addressing.

Regarding the original review, the response from Reviewer 1 was addressed satisfactorily. I feel my review is similar to Reviewer 3 in terms of its critique of the study’s assertion that additive advice is more burdensome or given too frequently and this issue still stands in the current manuscript. I agree with Reviewer 3’s consideration of mental health literacy as a potential confounder – this should be added in the discussion of the limitations that I recommend for this paper.

* TRANSPARENT PEER REVIEW: Communications Psychology uses a transparent peer review system. This means that we publish the editorial decision letters including Reviewers' comments to the authors and the author rebuttal letters online as a supplementary peer review file. However, on author request, confidential information and data can be removed from the published reviewer reports and rebuttal letters prior to publication. If your manuscript has been previously reviewed at another journal, those Reviewers' comments would not form part of the published peer review file.

If you experience problems in linking your ORCID, please contact the Platform Support Helpdesk.

Version 1:

Decision Letter:

Dear Dr Barry,

Thank you for submitting your revised manuscript titled "People overlook subtractive solutions to mental health problems." to Communications Psychology. We have given the paper our careful consideration and reviewed the preregistration for each study. It is our policy that all preregistered hypotheses/analyses must be reported except in the case of fundamental flaw or feasibility. In these cases, the deviation from the preregistration must be declared.

Currently, we are unable to locate the preregistered gender moderation analyses in Studies 4, 5, 6, and 7. For Study 3, we cannot locate the hypothesis/analysis related to karma as well as the moderation analysis by number of solutions already engaged in.

You can find our full policy on preregistration here: <https://www.nature.com/commpsychol/submit/preregistration>

We would therefore like to invite you to revise your manuscript to address these concerns.

We shall hope to receive your revised version as soon as you are able to complete the suggested revisions. If you anticipate a delay of more than four weeks, please let us know.

Please use the link below when you are prepared to resubmit.

Link Redacted

Thank you for your interest in Communications Psychology.

Best regards,
Jennifer Bellingtier

Jennifer Bellingtier, PhD
Senior Editor
Communications Psychology

Version 2:

Decision Letter:

Dear Dr Barry,

Your manuscript titled "People overlook subtractive solutions to mental health problems." has now been editorially reviewed, and I am delighted to say that we are happy, in principle, to publish a suitably revised version in Communications Psychology.

We therefore invite you to revise your paper one last time to address the remaining concerns of our reviewers and a list of editorial requests. At the same time we ask that you edit your manuscript to comply with our format requirements and to maximise the accessibility and therefore the impact of your work.

EDITORIAL REQUESTS:

SUBMISSION INFORMATION:

OPEN ACCESS:

* DATA AVAILABILITY:

Link Redacted

Best regards,

Jennifer Bellingtier

Jennifer Bellingtier, PhD
Senior Editor
Communications Psychology

Dear Dr. Bellingtier,

We are grateful to you for the chance to revise our manuscript even though both reviewers continue to have concerns about the manuscript. We address each of their points below and have revised the manuscript accordingly. We also note the editorial guidelines for this journal and have adapted the manuscript accordingly.

We thank you for your continued consideration.

With kindest regards,

The authors

Editor:

1. Editorially, we consider it important that the revised manuscript more thoroughly address the study limitations and situate the finding in the literature whilst staying close to what is shown by the data.

A: As is hopefully clear from our responses to the reviewers below we now acknowledge several limitations and potential sources of bias throughout the discussion section and have toned down our language accordingly.

2. Please rerun analyses including both vignette and participant as random effects.

A: As noted in our response to Reviewer 4, this is not always possible given the structure of the data but where possible we have done so. Although reviewer 4 does not mention including participant ID as a random effect, we had already done this throughout anyway.

3. Preregistration links for each study as well as dates of preregistration must be included in the Methods section. Any deviations from the preregistrations should be reported.

A: We have now listed every individual study link in the methods section and the date of pre-registration for each rather than the overall link to the project that appears in the data availability section.

Reviewer #2 (Remarks to the Author):

1. If you look at Figure 3, there is nothing magical about a p-value. The intuitive finding is that you have one pattern for friend and self - and a very different pattern for other. This is the difference that I would like to see reported and discussed.

A: We agree that more space could be given to interpreting the size of this effect. We now add the following to the discussion:

“There was evidence that who we are advising matters. Participants were more additive than subtractive when advising strangers and themselves but not a close friend (Study 7). However, although the contrast between additive and subtractive

advice was significant when advising oneself, the data in Figure 3 and the regression coefficient for this contrast indicate that this effect is small. Participants were significantly more subtractive to themselves than to strangers. Again, this could perhaps have been because people know more about themselves than the strangers, so it is easier to recommend subtractive advice. It is of note that this was within our study design that encourages subtractive responses by giving participants five spaces to give subtractive advice before they give additive advice. It is possible that, as with our other effects, this effect might be larger in other designs that allow open text advice giving. It is also of note, however, that the interaction between advice type and advice recipient was evident alongside the main effect of advice type. Irrespective of the advice recipient, the advice we give is markedly additive.”

2. Even if all participants were US based, information on their immigrant status (first and/or second generation) could still be analyzed to examine cultural differences.

A: Data on immigrant status are not available to us and in any case, we are reluctant to infer culture from immigrant status. Even if we were able to analyse immigrant status, it is unclear to us what such an analysis would actually tell us about culture per se. We certainly agree that culture could be a next step for future studies in this area but we would want such an analysis to examine how particular aspects of culture (e.g., independence/interdependence; differences in self-disclosure etc.) would influence the observed effects rather than proxies for culture that only allow us to say that there may be cultural differences but not what it is about culture that makes people differ in their responses. Unfortunately, such an analysis is not possible given the data available to us.

3. I believe that MHFA has a mostly additive agenda. It does not work, MHFA "aid recipients" are not better off (Forthal et al 2022; Richardson et al 2023, Cochrane). Still, MHFA is sold worldwide, a corrupt endeavor. If the authors agree with this interpretation, why do they not mention it? Their research should have a real-world impact.

A: We do not have a view on mental health first aid and we are not sure why we are being asked to take a view on it since it is not mentioned anywhere in the manuscript. We would add that the editorial guidelines of this journal prevent us from making intervention or policy recommendations where these were not directly tested. We feel that to comment on MHFA would violate this.

Reviewer #4 (Remarks to the Author):

1. While the claims that individuals are more likely to both give additive advice, and perceive it as beneficial compared to subtractive advice, are convincing, the paper does not establish that additive advice in the context of the study's eight experiments is a less effective, or more burdensome/stressful type of mental health advice than subtractive advice.

A: We agree that this is an important limitation and opportunity for future research. We now add the following in a new Limitations section within the Discussion:

“It is worth noting that the research presented here does not explore whether additive and subtractive advice differ in how beneficial or burdensome they are, once enacted. It remains possible that additive advice is indeed more beneficial than subtractive advice, however, intuitively it would seem that there are surely limits to how much additive advice one can follow.... Future research could explore this, perhaps within an Ecological Momentary Assessment design where advice benefit and burden are measured in the contexts in which advice adoption occurs.”

2. The methodological detail is fairly sound but there are several opportunities for bias in these methods that are not acknowledged by the manuscript. The development of the vignettes required binary coding of activities that may improve or worsen mental health (main manuscript, pg. 15), and the subsequent coding of the advice given by participants to these vignettes are also binary; as additive or subtractive advice (main manuscript, pg. 16). This assumes a clear distinction between these concepts, however in practice they can be mixed or ambiguous (e.g., “losing weight” might improve body image and mental health, or be a source of stress and poor body image). This could lead to coding being subjective, limiting replicability of this process.

A: We agree that these are potential sources of bias. However, as the reviewer remarks, the coding of activities that improve or worsen mental health was for the purposes of creating our vignettes in the early studies. As we state in the method, this was to deduce the activities that were most commonly used to manage mental health. As such, we only selected those activities that most of the 200 participants in the pilot study said affected their mental health. We are of the view that this reduced the ambiguity that the reviewer refers to. In any case, the reviewer is correct that this approach did introduce unexpected bias, as evidenced by our subsequent realisation that some of the activities were harmful by their presence (e.g., smoking) whereas others were harmful by the absence of something good (e.g., not going out). We analyse this in studies 6.5 and 7.

Regarding the binary nature of additive and subtractive coding. Our coding guides (all available on OSF) were based on those from the original study (Adam et al.) that also assumed advice giving was binary. It’s not clear to us how a piece of advice could be both additive and subtractive simultaneously. Losing weight is not an activity per se, it is an outcome that would require one to stop eating (subtractive), eating less unhealthy food (subtractive), eating more healthy food (additive) or exercise more (additive). We do believe that it is possible that activities can be more or less additive and subtractive or, put another way, for example, that some activities involve doing a lot more than others despite both being additive. We think this is worth mentioning as a limitation and so have added the following within the section where we talk about the real burden of activities (mentioned in response to comment 1 above):

“Also, our analyses, and those of other studies in this area^{14,15}, consider additive and subtractive advice as binary categories but it is possible that activities are more or less additive/subtractive, perhaps as a function of the effort required to enact them. Future research could explore this, perhaps within an Ecological Momentary Assessment design where advice benefit and burden are measured in the contexts in which advice adoption occurs.”

3. I also wonder, having looked at the vignettes used by the study in OSF (<https://osf.io/vucbr>), if they bias participants towards giving additive advice by including suggestions of additive activities as being desired by the individual in the vignette. For example, “Vignette 2 Sarah noticed that she has been feeling overwhelmingly tense, restless and agitated. She finds herself worrying excessively about her job security, completing her errands on time, her health and many other aspects of her daily routine. Sarah often ruminates on the past, obsessing over the significance of someone's actions. Sarah has expressed that her poor drinking habits have contributed to her current mental state. As well as, spending hours watching movies and tv shows. However, Sarah feels exercising daily could significantly improve her mental state. Engaging her creative side by painting could also lift her mood.” This vignette seems to suggest that Sarah believes that increasing her exercise and engaging in painting would improve her mental health, encouraging study participants to offer her this additive advice. This is perhaps a limitation to acknowledge in the discussion.

A: We agree with the reviewer. We included these details so that the participants had some basis upon which to offer advice based on the person's personal preferences and existing activities. Notably, when we adapt the vignettes for studies 7 and 8 we remove these additional details and still see the same effect. We are now clear about this in the Discussion in our new Limitations section:

“It is of note that, in the initial studies presented here, the vignettes included some details about activities that the person in the vignette preferred to do to help their mental health. This was intended to give participants some content upon which to base their advice. It is possible this may have biased participants towards recommending additive advice in line with these preferences. However, as can be seen in Figure 1, the average amount of additive advice given in the studies often exceeded the amount of these details that were present in the vignettes. In addition, these details were removed in Study 7, and they were not present in the Reddit posts analysed in Study 3, and yet the bias towards additive advice was present in both.”

4. Regarding the statistical analysis, I noticed that vignette ID was included as random effect in Study 4 but not in the other studies (supplementary material, pg. 3). The design of the other studies suggest that each participant is shown multiple vignettes, and advice is given on each vignette by multiple participants, so why was the vignette ID not included as a random effect in these other analyses? Adding vignette ID as a random effect to the LMMs for the other studies could also mitigate bias from the tendency for some vignettes to elicit more additive advice.

A: We thank the reviewer for catching this and we realise now that our description of this was inadequate. For clarity, it is not always possible for us to include vignette ID as a random effect.

For Study 1, there is only one vignette for each level of vignette gender and activity amount. For Study 3 and the analysis of reddit data we do include a random effect to capture the variance explained by the original post from which the comments we analyse are attached.

For Studies 2, 5 and 6 (and 6.5) we average across levels of each IV within each study. We re-ran the analyses removing the averaging and including random effects for vignette ID but the results are the same. We have chosen not to report these other models since they contradict our original pre-registration. If the editor and reviewer would prefer we are happy to report these alternate analyses instead.

For study 7, the analysis of self vs. other vs. friend does not require a vignette effect. In the analysis of harm within the vignettes for 'other' there is one vignette for each level of the IVs. However, we notice now that we failed to include a random effect for counterbalance (as we do in studies 6.5 and 8). We now include this in the R script on OSF but the results do not change so there is no correction to the main or supplementary results text.

Also, across all of the studies, we acknowledge that the descriptions of each study could be clearer to avoid this confusion. We have updated the description of each study to be clearer about the exact models and whether or not there was averaging.

6. I feel the claims are not entirely appropriately discussed in the context of the literature or findings. The study establishes through a range of studies that additive advice is more commonly given and rated as more beneficial. However, this is very distinct from the claims in the Abstract that "People feel that they are overburdened and yet we often tell them to do more. When helping people with their mental health, advice givers must balance what can be added to life with what needs to be removed" (main manuscript, pg. 2) and in the Discussion that "This has the potential to create a social context in which we are advising one another to always do more, leading to a worsening of perceived stress or burden" (main manuscript, pg. 9). The authors do not cite empirical literature linking additive advice to increased stress or burden. These claims are also not directly supported by the studies; indeed Study 4 participants judged the additive advice as more acceptable/feasible and were more likely to adopt it themselves.

A: The reviewer is correct and we agree that some of our statements go beyond what was found. The novelty of our studies are such that there is no literature we can lean on to support the claim that additive advice causes a sense of being overburdened even if this makes sense intuitively. As such, we agree that these claims should be toned down and that recommendations for future research should be given. In the section that the reviewer mentions, we now remove the mention of burden and say:

"We possess a reliable tendency to give additive advice for mental health problems. This bias has the potential to create a social context in which we are advising one another to always do more."

We also add the critique and recommendations for future research as previously mentioned:

"However, the research presented here does not explore whether additive and subtractive advice differ in how beneficial or burdensome they are, once enacted. It remains possible that additive advice is indeed more beneficial than subtractive"

advice, however, intuitively it would seem that there are surely limits to how much additive advice one can follow.... Future research could explore this, perhaps within an Ecological Momentary Assessment design where advice benefit and burden are measured in the contexts in which advice adoption occurs."

In addition, we change the sentences in the Conclusion:

"In a world where we already feel like we are time poor and doing too much, there is a sense that we must do yet more to cope with the sadnesses and anxieties of life. This might be compounded by advice to others that is predominantly telling them to do more. Future research should now explore the real-life burden of this additive advice bias."

We also removed the last sentence of the abstract so that that the last sentence now reads:

"People overlook subtractive advice when advising others and instead tell them to do more. Future research must explore the contribution of this additive advice bias to people's sense of being overwhelmed."

7. Furthermore, the additive and subtractive advice coded in the studies are sometimes semantically similar recommendations that may be functionally indistinguishable to the advisee; for example "I would suggest he improve his diet" is coded as additive while "she should cut her diet to be more healthy" is coded as subtractive (<https://osf.io/v3h9g>).

A: We agree that these examples are very similar though we would argue that improve versus cut feel distinct to us; that in study 5 onwards, the responses are coded by the advisor as either additive or subtractive so they can decide for themselves what they mean; and, if additive and subtractive advice were generally 'functionally indistinguishable' then we would not expect to see any difference in benefit or feasibility in study 4, 5, 6 or 7. That they are consistently significantly different in this respect suggests that for the most part they are viewed as quite distinct by the advisor and advisee. However, we acknowledge that none of this gets at limitations regarding *actual* rather than perceived benefit/burden and so we have made the other changes requested by the reviewer.

8. Thus, while the studies evidence a bias towards additive advice, the manuscript would benefit from a more cautious discussion of their implications. The conclusion that non-experts must consider "what we can do less of" may be true, but is not established by the studies and should be framed as a hypothesis for future research rather than a direct consequence of the findings.

My recommendations for the manuscript would be to acknowledge in the potential biases described above and evidence the idea of additive advice as burdensome with literature (or as a future research question) rather than as a finding of the research.

A: We hope that the changes listed above regarding the acknowledge of limitations, recommendations for future research, and toning down of language related to burden throughout alleviate these concerns.

9. There are some small spelling errors (“impove” should be “improve” on pg. 4, main manuscript) and some sentences are a bit confusing (“gave participants instructions about what additive and subtractive advice where and gave them practice giving each.” on pg. 6, main manuscript) that would be worth addressing.

A: We appreciate the reviewer catching these errors and drawing our attention to others. We have corrected throughout. In particular, “*...gave participants information about additive and subtractive advice, gave examples of each and gave them practice giving each.*”.

9. I agree with Reviewer 3’s consideration of mental health literacy as a potential confounder – this should be added in the discussion of the limitations that I recommend for this paper.

A: We now add a small section on this. We now add the following after the discussion about how people adapted their advice based on the kinds of activities the person in the vignette was suggested to be involved in (and after the sentence on GPT within this context):

“For human participants, mental health literacy – or knowledge about the factors that can be of positive and negative harm to mental health and the activities that might subsequently be beneficial to a person – is likely to be an important moderator of these effects that should be explored in future research.”

Dear Dr. Bellingtier,

We are grateful to you for the chance to revise our manuscript even though both reviewers continue to have concerns about the manuscript. We address each of their points below and have revised the manuscript accordingly. We also note the editorial guidelines for this journal and have adapted the manuscript accordingly.

In addition, we have also now corrected the inconsistencies with regards to reporting of pre-registrations and describe each below in response to your concerns.

We thank you for your continued consideration.

With kindest regards,

The authors

Editor:

1. Editorially, we consider it important that the revised manuscript more thoroughly address the study limitations and situate the finding in the literature whilst staying close to what is shown by the data.

A: As is hopefully clear from our responses to the reviewers below we now acknowledge several limitations and potential sources of bias throughout the discussion section and have toned down our language accordingly.

2. Please rerun analyses including both vignette and participant as random effects.

A: As noted in our response to Reviewer 4, this is not always possible given the structure of the data but where possible we have done so. Although reviewer 4 does not mention including participant ID as a random effect, we had already done this throughout anyway.

3. Preregistration links for each study as well as dates of preregistration must be included in the Methods section. Any deviations from the preregistrations should be reported.

A: We have now listed every individual study link in the methods section and the date of pre-registration for each rather than the overall link to the project that appears in the data availability section.

3a. We are unable to locate the preregistered gender moderation analyses in Studies 4, 5, 6, and 7.

A: We assume you mean vignette gender. We regret that we were unclear about this. This was described in earlier drafts of the manuscript but we appear to have removed it at some stage. In any case, to be clear that we are not hiding anything, this is included in our prior knowledge section of our Study 6.5 pre-registration. In short, we realised after Study 6/6.5 that vignette gender was confounded by 'harm':

“In our original analyses of each dataset we found effects of 'vignette gender' that reversed when study 6 counterbalanced the genders from study 5 (e.g., men showed bigger responses than women in study 5 and then women were bigger than men in study 6). After inspection of the vignettes, we realised that one way in which they differ besides the gender we had assigned them was in the number of positive vs. negative harms.”

In Studies 4, 5 and 6, we found significant effects of 'vignette gender' but these were really effects of harm, given the confound. To account for this variability, we continue to include the vignette gender variable in our models but, to avoid misleading readers into thinking there were vignette gender effects, we do not report these data. As such, we do not draw any conclusions about vignette gender from studies 4, 5 and 6. We are now clear about this in the introduction to Study 4:

“Analyses of Study 4 data were pre-registered to include an analysis of vignette gender and the analyses reported here included this variable within the models. However, following Study 6.5, it was clear these analyses were confounded by the harm type described within the vignettes (see Study 6.5). We continue to include vignette gender within the models because of our pre-registered plan and to account for the variance explained by this confounding variable, but we do not present the effect of vignette gender so as to avoid suggesting that there were significant effects of vignette gender. Removal of the vignette gender variable from the models does not change any of the significant effects.”

We reiterate this briefly in the description of studies 5 and 6. Regarding Study 7, since in this study there was proper counterbalancing of vignette gender and harm type, we are able to report on vignette gender. However, there were no significant effects of it and so we present these analyses in the supplement. This is mentioned in the Discussion already.

3b. For Study 3, we cannot locate the hypothesis/analysis related to karma as well as the moderation analysis by number of solutions already engaged in.

A: We regret these inconsistencies also. The analysis of number of activities was conducted but was only reported in the supplement. We have moved this to the main results section now. Regarding comment Karma, as you can see from the updates to the OSF we had difficulty getting these data due to changes in the Reddit API during the life of the project. When we were eventually able to get the data, it proved fruitless, since there was so little comment karma in the responses that were analysed. We now include a note on this in the main text after the pre-registration link for Study 3.

“Our pre-registered analysis plan included an analysis of whether responses that were given more additive versus subtractive were assigned more or less comment karma/upvotes by other Reddit users. However, after scraping these data, these analyses were not possible given floor effects produced by the low amount of karma given to responses which prevented our models from converging.”

Reviewer #2 (Remarks to the Author):

1. If you look at Figure 3, there is nothing magical about a p-value. The intuitive finding is that you have one pattern for friend and self - and a very different pattern for other. This is the difference that I would like to see reported and discussed.

A: We agree that more space could be given to interpreting the size of this effect. We now add the following to the discussion:

“There was evidence that who we are advising matters. Participants were more additive than subtractive when advising strangers and themselves but not a close friend (Study 7). However, although the contrast between additive and subtractive advice was significant when advising oneself, the data in Figure 3 and the regression coefficient for this contrast indicate that this effect is small. Participants were significantly more subtractive to themselves than to strangers. Again, this could perhaps have been because people know more about themselves than the strangers, so it is easier to recommend subtractive advice. It is of note that this was within our study design that encourages subtractive responses by giving participants five spaces to give subtractive advice before they give additive advice. It is possible that, as with our other effects, this effect might be larger in other designs that allow open text advice giving. It is also of note, however, that the interaction between advice type and advice recipient was evident alongside the main effect of advice type. Irrespective of the advice recipient, the advice we give is markedly additive.”

2. Even if all participants were US based, information on their immigrant status (first and/or second generation) could still be analyzed to examine cultural differences.

A: Data on immigrant status are not available to us and in any case, we are reluctant to infer culture from immigrant status. Even if we were able to analyse immigrant status, it is unclear to us what such an analysis would actually tell us about culture per se. We certainly agree that culture could be a next step for future studies in this area but we would want such an analysis to examine how particular aspects of culture (e.g., independence/interdependence; differences in self-disclosure etc.) would influence the observed effects rather than proxies for culture that only allow us to say that there may be cultural differences but not what it is about culture that makes people differ in their responses. Unfortunately, such an analysis is not possible given the data available to us.

3. I believe that MHFA has a mostly additive agenda. It does not work, MHFA "aid recipients" are not better off (Forthal et al 2022; Richardson et al 2023, Cochrane). Still, MHFA is sold worldwide, a corrupt endeavor. If the authors agree with this interpretation, why do they not mention it? Their research should have a real-world impact.

A: We do not have a view on mental health first aid and we are not sure why we are being asked to take a view on it since it is not mentioned anywhere in the manuscript. We would add that the editorial guidelines of this journal prevent us from making intervention or policy recommendations where these were not directly tested. We feel that to comment on MHFA would violate this.

Reviewer #4 (Remarks to the Author):

1. While the claims that individuals are more likely to both give additive advice, and perceive it as beneficial compared to subtractive advice, are convincing, the paper does not establish that additive advice in the context of the study's eight experiments is a less effective, or more burdensome/stressful type of mental health advice than subtractive advice.

A: We agree that this is an important limitation and opportunity for future research. We now add the following in a new Limitations section within the Discussion:

"It is worth noting that the research presented here does not explore whether additive and subtractive advice differ in how beneficial or burdensome they are, once enacted. It remains possible that additive advice is indeed more beneficial than subtractive advice, however, intuitively it would seem that there are surely limits to how much additive advice one can follow.... Future research could explore this, perhaps within an Ecological Momentary Assessment design where advice benefit and burden are measured in the contexts in which advice adoption occurs."

2. The methodological detail is fairly sound but there are several opportunities for bias in these methods that are not acknowledged by the manuscript. The development of the vignettes required binary coding of activities that may improve or worsen mental health (main manuscript, pg. 15), and the subsequent coding of the advice given by participants to these vignettes are also binary; as additive or subtractive advice (main manuscript, pg. 16). This assumes a clear distinction between these concepts, however in practice they can be mixed or ambiguous (e.g., "losing weight" might improve body image and mental health, or be a source of stress and poor body image). This could lead to coding being subjective, limiting replicability of this process.

A: We agree that these are potential sources of bias. However, as the reviewer remarks, the coding of activities that improve or worsen mental health was for the purposes of creating our vignettes in the early studies. As we state in the method, this was to deduce the activities that were most commonly used to manage mental health. As such, we only selected those activities that most of the 200 participants in the pilot study said affected their mental health. We are of the view that this reduced the ambiguity that the reviewer refers to. In any case, the reviewer is correct that this approach did introduce unexpected bias, as evidenced by our subsequent realisation that some of the activities were harmful by their presence (e.g., smoking) whereas others were harmful by the absence of something good (e.g., not going out). We analyse this in studies 6.5 and 7.

Regarding the binary nature of additive and subtractive coding. Our coding guides (all available on OSF) were based on those from the original study (Adam et al.) that also assumed advice giving was binary. It's not clear to us how a piece of advice could be both additive and subtractive simultaneously. Losing weight is not an activity per se, it is an outcome that would require one to stop eating (subtractive), eating less unhealthy food (subtractive), eating more healthy food (additive) or exercise more (additive). We do believe that it is possible that activities can be more or less additive and subtractive or, put another way, for example, that some activities involve doing a lot more than others despite both being additive. We think this is worth mentioning as a limitation and so have added the following within the section

where we talk about the real burden of activities (mentioned in response to comment 1 above):

“Also, our analyses, and those of other studies in this area^{14,15}, consider additive and subtractive advice as binary categories but it is possible that activities are more or less additive/subtractive, perhaps as a function of the effort required to enact them. Future research could explore this, perhaps within an Ecological Momentary Assessment design where advice benefit and burden are measured in the contexts in which advice adoption occurs.”

3. I also wonder, having looked at the vignettes used by the study in OSF (<https://osf.io/vucbr>), if they bias participants towards giving additive advice by including suggestions of additive activities as being desired by the individual in the vignette. For example, “Vignette 2 Sarah noticed that she has been feeling overwhelmingly tense, restless and agitated. She finds herself worrying excessively about her job security, completing her errands on time, her health and many other aspects of her daily routine. Sarah often ruminates on the past, obsessing over the significance of someone's actions. Sarah has expressed that her poor drinking habits have contributed to her current mental state. As well as, spending hours watching movies and tv shows. However, Sarah feels exercising daily could significantly improve her mental state. Engaging her creative side by painting could also lift her mood.” This vignette seems to suggest that Sarah believes that increasing her exercise and engaging in painting would improve her mental health, encouraging study participants to offer her this additive advice. This is perhaps a limitation to acknowledge in the discussion.

A: We agree with the reviewer. We included these details so that the participants had some basis upon which to offer advice based on the person's personal preferences and existing activities. Notably, when we adapt the vignettes for studies 7 and 8 we remove these additional details and still see the same effect. We are now clear about this in the Discussion in our new Limitations section:

“It is of note that, in the initial studies presented here, the vignettes included some details about activities that the person in the vignette preferred to do to help their mental health. This was intended to give participants some content upon which to base their advice. It is possible this may have biased participants towards recommending additive advice in line with these preferences. However, as can be seen in Figure 1, the average amount of additive advice given in the studies often exceeded the amount of these details that were present in the vignettes. In addition, these details were removed in Study 7, and they were not present in the Reddit posts analysed in Study 3, and yet the bias towards additive advice was present in both.”

4. Regarding the statistical analysis, I noticed that vignette ID was included as random effect in Study 4 but not in the other studies (supplementary material, pg. 3). The design of the other studies suggest that each participant is shown multiple vignettes, and advice is given on each vignette by multiple participants, so why was the vignette ID not included as a random effect in these other analyses? Adding vignette ID as a random effect to the LMMs for the other studies could also mitigate bias from the tendency for some vignettes to elicit more additive advice.

A: We thank the reviewer for catching this and we realise now that our description of this was inadequate. For clarity, it is not always possible for us to include vignette ID as a random effect.

For Study 1, there is only one vignette for each level of vignette gender and activity amount. For Study 3 and the analysis of reddit data we do include a random effect to capture the variance explained by the original post from which the comments we analyse are attached.

For Studies 2, 5 and 6 (and 6.5) we average across levels of each IV within each study. We re-ran the analyses removing the averaging and including random effects for vignette ID but the results are the same. We have chosen not to report these other models since they contradict our original pre-registration. If the editor and reviewer would prefer we are happy to report these alternate analyses instead.

For study 7, the analysis of self vs. other vs. friend does not require a vignette effect. In the analysis of harm within the vignettes for 'other' there is one vignette for each level of the IVs. However, we notice now that we failed to include a random effect for counterbalance (as we do in studies 6.5 and 8). We now include this in the R script on OSF but the results do not change so there is no correction to the main or supplementary results text.

Also, across all of the studies, we acknowledge that the descriptions of each study could be clearer to avoid this confusion. We have updated the description of each study to be clearer about the exact models and whether or not there was averaging.

6. I feel the claims are not entirely appropriately discussed in the context of the literature or findings. The study establishes through a range of studies that additive advice is more commonly given and rated as more beneficial. However, this is very distinct from the claims in the Abstract that "People feel that they are overburdened and yet we often tell them to do more. When helping people with their mental health, advice givers must balance what can be added to life with what needs to be removed" (main manuscript, pg. 2) and in the Discussion that "This has the potential to create a social context in which we are advising one another to always do more, leading to a worsening of perceived stress or burden" (main manuscript, pg. 9). The authors do not cite empirical literature linking additive advice to increased stress or burden. These claims are also not directly supported by the studies; indeed Study 4 participants judged the additive advice as more acceptable/feasible and were more likely to adopt it themselves.

A: The reviewer is correct and we agree that some of our statements go beyond what was found. The novelty of our studies are such that there is no literature we can lean on to support the claim that additive advice causes a sense of being overburdened even if this makes sense intuitively. As such, we agree that these claims should be toned down and that recommendations for future research should be given. In the section that the reviewer mentions, we now remove the mention of burden and say:

“We possess a reliable tendency to give additive advice for mental health problems. This bias has the potential to create a social context in which we are advising one another to always do more.”

We also add the critique and recommendations for future research as previously mentioned:

“However, the research presented here does not explore whether additive and subtractive advice differ in how beneficial or burdensome they are, once enacted. It remains possible that additive advice is indeed more beneficial than subtractive advice, however, intuitively it would seem that there are surely limits to how much additive advice one can follow.... Future research could explore this, perhaps within an Ecological Momentary Assessment design where advice benefit and burden are measured in the contexts in which advice adoption occurs.”

In addition, we change the sentences in the Conclusion:

“In a world where we already feel like we are time poor and doing too much, there is a sense that we must do yet more to cope with the sadnesses and anxieties of life. This might be compounded by advice to others that is predominantly telling them to do more. Future research should now explore the real-life burden of this additive advice bias.”

We also removed the last sentence of the abstract so that that the last sentence now reads:

“People overlook subtractive advice when advising others and instead tell them to do more. Future research must explore the contribution of this additive advice bias to people’s sense of being overwhelmed.”

7. Furthermore, the additive and subtractive advice coded in the studies are sometimes semantically similar recommendations that may be functionally indistinguishable to the advisee; for example “I would suggest he improve his diet” is coded as additive while “she should cut her diet to be more healthy” is coded as subtractive (<https://osf.io/v3h9g>).

A: We agree that these examples are very similar though we would argue that improve versus cut feel distinct to us; that in study 5 onwards, the responses are coded by the advisor as either additive or subtractive so they can decide for themselves what they mean; and, if additive and subtractive advice were generally ‘functionally indistinguishable’ then we would not expect to see any difference in benefit or feasibility in study 4, 5, 6 or 7. That they are consistently significantly different in this respect suggests that for the most part they are viewed as quite distinct by the advisor and advisee. However, we acknowledge that none of this gets at limitations regarding *actual* rather than perceived benefit/burden and so we have made the other changes requested by the reviewer.

8. Thus, while the studies evidence a bias towards additive advice, the manuscript would benefit from a more cautious discussion of their implications. The conclusion that non-experts must consider “what we can do less of” may be true, but is not

established by the studies and should be framed as a hypothesis for future research rather than a direct consequence of the findings.

My recommendations for the manuscript would be to acknowledge in the potential biases described above and evidence the idea of additive advice as burdensome with literature (or as a future research question) rather than as a finding of the research.

A: We hope that the changes listed above regarding the acknowledge of limitations, recommendations for future research, and toning down of language related to burden throughout alleviate these concerns.

9. There are some small spelling errors (“improve” should be “improve” on pg. 4, main manuscript) and some sentences are a bit confusing (“gave participants instructions about what additive and subtractive advice where and gave them practice giving each.” on pg. 6, main manuscript) that would be worth addressing.

A: We appreciate the reviewer catching these errors and drawing our attention to others. We have corrected throughout. In particular, “...*gave participants information about additive and subtractive advice, gave examples of each and gave them practice giving each.*”.

9. I agree with Reviewer 3’s consideration of mental health literacy as a potential confounder – this should be added in the discussion of the limitations that I recommend for this paper.

A: We now add a small section on this. We now add the following after the discussion about how people adapted their advice based on the kinds of activities the person in the vignette was suggested to be involved in (and after the sentence on GPT within this context):

“For human participants, mental health literacy – or knowledge about the factors that can be of positive and negative harm to mental health and the activities that might subsequently be beneficial to a person – is likely to be an important moderator of these effects that should be explored in future research.”